# Optimization of Physical Activation Process by CO_2_ for Activated Carbon Preparation from Honduras Mahogany Pod Husk

**DOI:** 10.3390/ma16196558

**Published:** 2023-10-05

**Authors:** Chi-Hung Tsai, Wen-Tien Tsai

**Affiliations:** 1Department of Resources Engineering, National Cheng Kung University, Tainan 701, Taiwan; ap29fp@gmail.com; 2Graduate Institute of Bioresources, National Pingtung University of Science and Technology, Pingtung 912, Taiwan

**Keywords:** Honduras Mahogany, seed husk, physical activation, activated carbon, pore analysis, process optimization, resource reuse

## Abstract

In this work, the Honduras Mahogany (*Swietenia macropnylla* King, SMK) seed husk was used as a novel biomass resource for producing activated carbon by physical activation. The texture characteristics and chemical characterization of resulting products were investigated in correlation with the process parameters. Based on the thermochemical properties of the SMK biomass, the process conditions were set to a rate of about 10 °C/min under nitrogen (N_2_) flow of 500 cm^3^/min heated to 500 °C, then switched to carbon dioxide (CO_2_) flow of 100 cm^3^/min in the specified activation conditions (i.e., temperature of 700–850 °C for holding times of 0–60 min). Our findings showed that the texture characteristics (i.e., surface area and pore volume) increased with an activation temperature increase from 700 to 800 °C for a holding time of 30 min but gradually decreased as the temperature increased thereafter. Similarly, the texture characteristics also indicated an increasing trend with the residence time extending from 0 min to 30 min but slightly decreased as the time was extended to 60 min. Therefore, the optimal activation conditions for producing SMK-based activated carbon should be set at 800 °C for a holding time of 30 min to obtain the maximal texture characteristics (i.e., BET surface area of 966 m^2^/g and total pore volume of 0.43 cm^3^/g). On the other hand, the chemical characteristics were analyzed by energy dispersive X-ray spectroscopy (EDS) and Fourier Transform infrared spectroscopy (FTIR), showing oxygen complexes contained on the hydrophilic surface of the resulting activated carbon.

## 1. Introduction

Due to its excellent texture characteristics, activated carbon has been widely used in a variety of industrial and environmental applications, such as gas-phase/liquid-phase adsorbent for purification/remediation [1,2]. Due to the presence of surface oxygen complexes, liquid-phase (mainly water) adsorption for removing organic/inorganic pollutants is more common than gas-phase adsorption. Therefore, activated carbon sometimes acts as an ion-exchange material. On the other hand, a novel application of activated carbon derived from biomass has been used in electrochemical energy storage devices in recent years [3,4,5]. In industrial/commercial production, the precursors for producing activated carbon mainly include coal and hard husk (e.g., coconut shell) [6], but the resulting carbon products are relatively expensive. In this regard, it is necessary to find a new biomass precursor for the production of microporous carbon materials. Therefore, a variety of lignocellulosic residues for producing activated carbon have been recently reviewed in the literature [7,8,9,10].

*Swietenia macropnylla* King (SMK) belongs to the family Meliaceae, which is commonly known as mahogany. It is native to the tropical regions of America, spreading from southern Mexico to the North of Brazil [11]. In Taiwan, SMK has been extensively planted in the plain area to exploit it as available wood material for making furniture and other advanced wood products. Its seeds (or pods) often fall to the ground after ripening or blowing by the wind. Concerning the utilization of the SMK seed pod, it can be reused as a source of natural colorants, which may be important dyes in the fabric texture [12]. In addition, the SMK seeds can be collected to be reused as an energy source due to their lignocellulosic constituents [13]. In this work, a novel biomass Honduras Mahogany (*Swietenia macropnylla* King, SMK) seed husk (pod) was tested as a starting material in the production of activated carbon.

Regarding the production process of activated carbon, its porosities are not sufficiently developed during the carbonization phase. Therefore, the physical (or thermal) activation by carbon dioxide (CO_2_) or steam gas flow at higher temperatures (above 700 °C) was performed to create more pores, causing higher texture characteristics such as surface area. Although the chemical activation can be operated at lower temperatures, this process may generate environmental pollution problems due to the release of toxic activation components (e.g., zinc and phosphorus) from the water-washing streams. As a consequence, the commercial process for manufacturing activated carbon commonly adopted the physical activation method [3].

Based on the survey of the academic database, the SMK biomass was not yet used as a precursor for producing activated carbon in the literature. Therefore, the thermochemical properties of the precursor were determined in this work for the first time. Using an environment friendly process, the physical activation experiments using CO_2_ gas flow were adopted to produce the activated carbon products from the dried SMK biomass at different activation temperatures (i.e., 700–850 °C) for various holding times (i.e., 0, 15, 30, 45, and 60 min). Furthermore, the texture characteristics were obtained by the measurements of N_2_ adsorption–desorption isotherms at −196 °C. The textural characteristics were observed by scanning electron microscopy (SEM). In order to characterize the oxygen-containing complexes on the surface, the chemical characteristics were obtained by energy dispersive X-ray spectroscopy (EDS) and the Fourier Transform infrared spectroscopy (FTIR).

## 2. Materials and Methods

### 2.1. Materials

The starting precursor for producing activated carbon (i.e., SMK) was collected from the university campus (National Pingtung University of Science and Technology, Pingtung, Taiwan). The sun-dried biomass was cut into small pieces, which were further pulverized by a crusher and sieved to obtain samples with the particle sizes from mesh No. 20 (opening size 0.841 mm, passed) to mesh No. 40 (opening size 0.420 mm, retained). Therefore, average particle size was about 0.63 mm. This biomass sample (as-received sample) was first used to measure its proximate analysis. Figure 1 shows the SMK seeds, flaky husk, and fine granules. Prior to the thermochemical property analyses and the carbonization–activation experiments, the SMK sample was dried in an air-circulation oven for 24 h at 105 °C.

### 2.2. Thermochemical Characteristics Analysis of SMK

In this work, the proximate analysis, calorific value, and thermogravimetric analysis (TGA) of the SMK sample were measured to determine its thermochemical properties with relevance to the potential for producing activated carbon. The American Society for Testing and Materials (ASTM) standard methods were adopted to obtain its contents of moisture, volatile matter (VM), and ash as wt%. The fixed carbon content was then determined by deducting the combined values of moisture, ash, and VM from 100. The calorific value of the dried SMK was measured by a bomb calorimeter (CALORIMETER ASSY 6200; Parr Co., Moline, IL, USA), and about 0.3 g of the sample was used for each analysis. The calorific value is related to the carbon content of the dried biomass sample. To understand the thermal decomposition behavior of the dried SMK sample, the TGA instrument (TGA-51; Shimadzu Co., Kyoto, Japan) was used to obtain the data on the weight loss percentage vs. temperature. Under the condition of nitrogen gas flow of 50 cm^3^/min, the heating profile was set from room temperature to 1000 °C at a rate of 10 °C/min.

### 2.3. Pyrolysis–Activation Experiments

The pyrolysis–activation experiments were described in previous studies [14,15,16,17,18]. In this work, about 5 g of dried SMK was fed into the physical activation system, which was performed at a heating rate of about 10 °C/min. The process conditions are described below. In the first stage (pyrolysis), the system temperature increased from 25 °C (room temperature) to 500 °C under nitrogen (N_2_) gas (flow rate of 500 cm^3^/min). In the second stage (activation), carbon dioxide (CO_2_) gas (flow rate of 100 cm^3^/min) was applied while stopping N_2_ gas and then heated to the prescribed process conditions (i.e., activation temperature of 700–850 °C and holding time of 0–60 min). The resulting activated carbon products were coded as SMK-temperature-time. For example, SMK-800-30 refers to the activated carbon produced at activation temperature of 800 °C for holding time of 30 min. In order to compare the pore properties of the resulting activated carbon with those of a carbon product without CO_2_ activation, an SMK-based biochar (noted as SMK-800-30-biochar) was produced at 800 °C for holding time of 30 min under nitrogen gas.

### 2.4. Characterization Analysis of Resulting Activated Carbon

The texture characteristics of the resulting activated carbon products were analyzed by the adsorption–desorption isotherms of high purity nitrogen (N_2_) at liquid nitrogen temperature (77 K, or −196 °C) using the ASAP 2020 Plus instrument (Micromeritics Co., Norcross, GA, USA). The data included surface area, pore size (or diameter), pore volume, and pore size distribution. Prior to this analysis, the carbon samples (about 0.2 g, dried at 100 °C) were degassed in a vacuum (≤1.33 Pa) at 200 °C for several hours. In this work, the data on surface area were obtained by the Brunauer–Emmett–Teller (BET) equation [19,20,21], based on the relative pressure (P/P_0_) values ranging from 0.05 to 0.15. It should be noted that the data on the BET surface area must meet the Rouquerol criteria (i.e., the value of the parameter C exceeds zero by denoting the positive y-intercept of the linear region), ensuring the validity of the BET model analysis for the microporous SMK-based activated carbon [22]. Herein, the pore volume referred to total pore volume, which was estimated by the assumption that all micropores (pore diameter or width: <2.0 nm) and mesopores (pore diameter or width: 2.0–50.0 nm) were filled by the nitrogen molecules at a condensed liquid state. Using the value of the adsorbed nitrogen amount (mol/g, or converted into g/g) at saturated relative pressure (i.e., 0.99), the property can be obtained by dividing the nitrogen liquid density (i.e., 0.806 g/cm^3^). With the model of slit geometry, the average pore diameter for slit pores can thus be calculated from the BET surface area and the total pore volume [19,20,21]. Assuming the capillary condensation of the liquid nitrogen within the pores, the Barrett–Joyner–Halenda (BJH) method was adopted to calculate the mesopore size distribution of the activated carbon [19,20,21]. Due to the fact that there are different branches in the adsorption and desorption isotherms, the desorption branch was employed for the calculation of mesopore size distribution. To depict the pore size distribution of less than 2 nm, the 2D-NLDFT-HS model was used in this work, which was based on the slit-cylinder pore shape boundary fixed at 2 nm [23]. In addition, scanning electron microscopy (SEM) (S-3000N; Hitachi Co., Tokyo, Japan) was employed to observe the porous surface morphology of the resulting activated carbon at an accelerating voltage of 15.0 kV. Prior to the SEM analysis, the dried sample was coated with a thin gold film using an ion sputter (E1010; Hitachi Co., Tokyo, Japan) because the surface of activated carbon is not electrically conductive.

Regarding the chemical characterization of the resulting activated carbon, the Fourier Transform infrared spectroscopy (FTIR) instrument (FT/IR-4600, JASCO Co., Tokyo, Japan) was applied to identify the surface chemical groups of the activated carbon samples. The spectra were recorded within the wavenumber range of 4000–400 cm^−1^. Before the FTIR analysis, potassium bromide (KBr, IR spectroscopy grade) was mixed with each activated carbon sample in an about one-to-ten ratio (1:10) and pelletized with a hydraulic press. In addition, the elemental compositions on the surface of the resulting activated carbon products and their starting precursor (i.e., SMK) were scanned using energy-dispersive X-ray spectroscopy (EDS). When observing the surface texture by SEM, the EDS instrument (7021-H; HORIBA Co., Osaka, Japan) was employed to semi-quantify their contents of carbon, oxygen, and other inorganic elements [24].

## 3. Results and Discussion

### 3.1. Thermochemical Properties of SMK

The proximate analysis of the as-received SMK biomass showed values of 7.85 ± 0.49 wt% for moisture, 73.38 ± 2.33 wt% for volatile matter, 4.63 ± 0.25 wt% for ash and 14.14 wt% for fixed carbon, which were measured in triplicate. This biomass had a moderate ash content, thus causing the calorific value (17.57 ± 0.18 MJ/kg, dry basis) to range from that of rice residues to that of woody biomass [25,26,27]. Based on the results of the energy dispersive X-ray spectroscopy (EDS) analysis, the main elemental contents of the SMK biomass were carbon (C, 54.7 wt%), oxygen (O, 40.3 wt%), and potassium (K, 3.9 wt%). The minor elements included calcium (Ca) and sodium (Na). Furthermore, the thermogravimetric analysis (TGA) and its derivative thermogravimetry (DTG) curves of the SMK biomass are shown in Figure 2. It can be seen that about 60 wt% of the sample was thermally decomposed at a pyrolysis temperature of 250 to 400 °C. The maximal weight loss rate occurred at around 320 °C, which should be attributed to the thermal decomposition of hemicellulose/cellulose constituents [28,29,30]. Using the TGA/DTG results, the pyrolysis condition was performed at a temperature (i.e., 500 °C) during the first process stage to produce carbon-rich biochar, which was subsequently activated at higher temperatures (i.e., 700–850 °C) in the second process stage.

### 3.2. Texture Characteristics of Resulting Activated Carbon

As depicted in Figure 2, the weight loss of the SMK sample decreased when the pyrolysis temperature gradually increased from 400 °C to 1000 °C. In this work, the mass yield at higher activation temperatures (>850 °C) was close to zero, implying the burn-off without activated carbon produced. Table 1 lists the mass yields of the resulting activated carbon products produced at 700–835 °C for a holding time of 30 min and at 800 °C for holding times of 0–60 min. As expected, the data on the mass yield indicated a decreasing trend at higher temperatures and longer holding times because of more intense activation (or gasification) reactions. In addition, the weight loss rates in the pyrolysis physical activation process indicated a stable variation, which was similar to those in Figure 2. From the viewpoint of cost-effectiveness and energy consumption, the low mass yield was not preferred for producing activated carbon products at higher activation temperatures, even if they had larger pore properties. In this regard, the activation temperature of 800 °C was fixed to study the effect of residence time (0–60 min) on the texture characteristics of the resulting activated carbon.

In the physical activation process, the final process temperature may be the most important parameter for determining the texture characteristics of the activated carbon prepared from the biomass precursor [2]. In addition, the residence (or holding) time at the final activation temperature is another process parameter that affects the characteristics of activated carbon. The data in Table 1 summarize the main texture characteristics (i.e., BET surface area, total pore volume, and average pore diameter thus estimated) of the resulting activated carbon products produced by the combined conditions under the activation temperature (700–850 °C) and residence time (0–60 min). Figure 3 shows the N_2_ adsorption–desorption isotherms (i.e., −196 °C) of the resulting products prepared under various activation conditions. Figure 4 further depicts the pore size distributions based on the BJH method. Based on the results of Table 1, Figure 3, and Figure 4, the significant findings were addressed as follows:It was found that the mass yields of activated carbon produced at higher activation temperatures and larger residence times decreased continuously, causing gradual shrinkage and burn-off. Because of more CO_2_ gasification (activation) reactions occurring at elevated temperatures, the resulting activated carbon product was not produced or burned out at above 850 °C for a holding time of 30 min. Therefore, the lower activation temperatures at 825 °C and 835 °C were used to investigate the variations in the mass yields and texture characteristics. On the other hand, the texture characteristics of activated carbon increased as the activation temperature increased from 700 to 850 °C, giving more pore formation and the increment of the mesopore surface area (mesoporosity). Eventually, the effects of activation (or gasification) on the texture characteristics reached the optimal process conditions, where the value of the BET surface area (or total pore volume) was close to the maximum. As listed in Table 1, the main texture characteristic (the BET surface area) increased with the activation temperature increase from 700 to 800 °C for a holding time of 30 min but gradually decreased as the temperature increased thereafter from the BET surface area of 966 m^2^/g (SMK-800-30) to 791 m^2^/g (SMK-825-30) and 475 m^2^/g (SMK-835-30). More consistently, the total pore volume indicated a similar trend, suggesting more micropores formed at higher activation temperatures of 700 °C to 800 °C. Based on these texture characteristics, the activation temperatures ranging from 750 to 800 °C can be connected with the determining process conditions for the most surface area and pore volume gained in this work.Also indicated in Table 1, the residence time may be another determining process parameter in activated carbon production. The variations in the main texture characteristics were similar to those mentioned above. For example, the BET surface area increased with extending residence time from 0 min to 30 min but slightly decreased as the time extended thereafter from the maximal BET surface area (SMK-800-30) to 909 m^2^/g (SMK-800-45) and 609 m^2^/g (SMK-835-60). Therefore, the optimal activation conditions for producing SMK-based activated carbon should be fixed at 800 °C for a holding time of 30 min, having the maximal texture characteristics (i.e., BET surface area of 966 m^2^/g and total pore volume of 0.43 cm^3^/g).Although the mass yield of the resulting biochar (SMK-800-30-biochar) was higher than that of the resulting activated carbon (SMK-800-30), its texture characteristics were significantly lower than those of the latter. The CO_2_ gas played a vital role in the pore enhancement during the activation or gasification reaction (i.e., carbon and carbon dioxide reaction), causing the removal of surface oxygen complexes and more pore formation [2].As shown in Figure 3, the resulting activated carbon products are characteristic of microporous features, thus displaying the Type I isotherms based on the classification by the International Union of Pure and Applied Chemistry (IUPAC) [19,20,21]. In addition, the average pore width for slit pores (W_p_) was close to 4.0 nm, based on the model of slit geometry. Further, there were slight hysteresis loops (Type VI isotherms) starting from approximately 0.42 of relative pressure in the desorption branch of the isotherms, which were associated with the development of mesopores. It should be noted that the pore size distributions obtained by the BJH method using the desorption branch data were not correct because the carbon materials are fundamentally microporous and do not present a considerable number of mesopores.By analyzing the adsorption–desorption isotherm data of a series of CO_2_-activated SMK, Figure 4 further depicts the micropore size distribution in the optimal activated carbon (i.e., SMK-800-30) using the 2D-NLDFT-HS model for a more accurate description of the textural characteristics [23]. The resulting activated carbon was a microporous material indeed and showed a significant micropore at 0.6 nm. In addition, these micropores should be slit-based shape as compared to the cylinder-based shape in the mesopores.As mentioned above, the optimal activated carbon product was the SMK-800-30, which had a BET surface area of 966 m^2^/g and a total pore volume of 0.43 cm^3^/g. In this work, scanning electron microscopy (SEM) was used to observe its porous texture on the surface using two magnifications (i.e., ×1000 and ×3000). As illustrated in the left image of Figure 5, it exhibited a smooth and rigid surface due to its carbon matrix derived from the vascular structure in the lignocellulosic biomass (i.e., SMK). When the SEM image was zoomed in at a higher magnification (i.e., ×3000, the right image of Figure 5), the activated carbon product displayed a more porous texture on its surface.

**Figure 3 materials-16-06558-f003:**
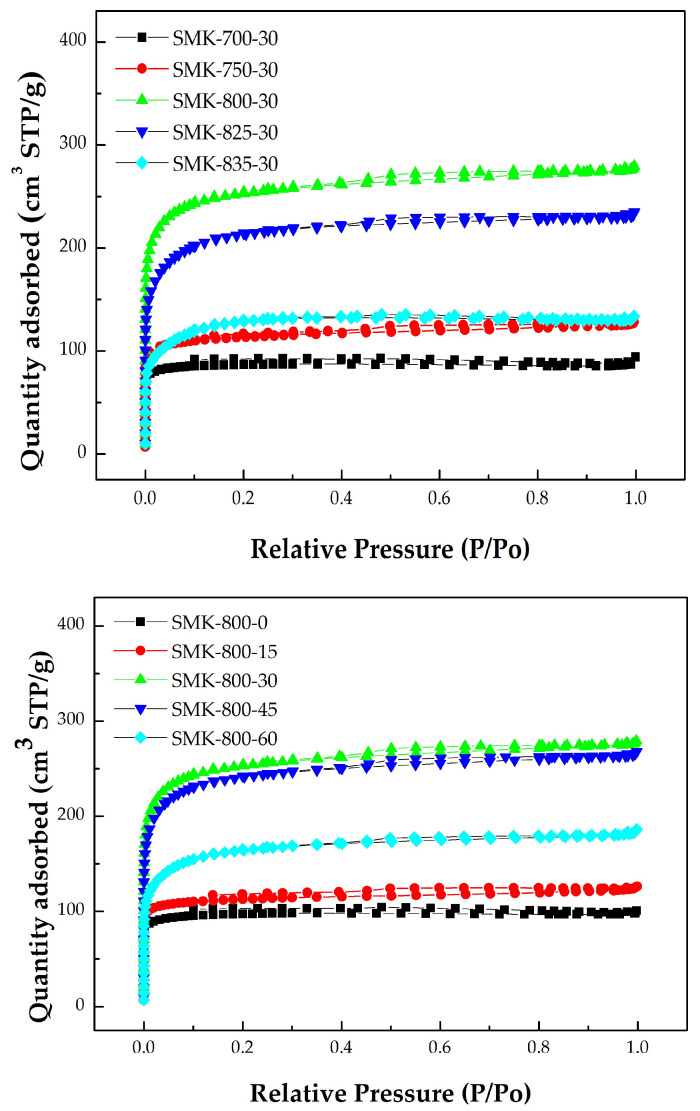
N_2_ adsorption–desorption isotherms of activated carbon produced from the biomass precursor (i.e., SMK) at various activation conditions (Upper figure: different temperatures at residence time of 30 min; Lower figure: different residence times at temperature of 800 °C).

**Figure 4 materials-16-06558-f004:**
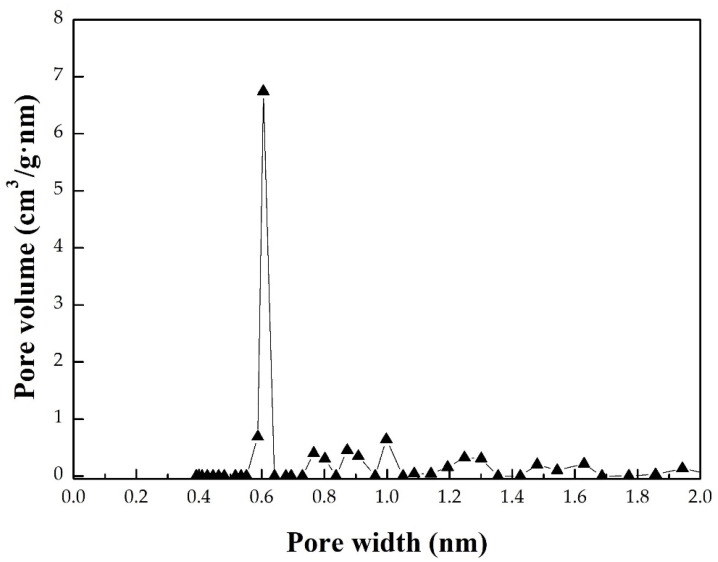
Micropore size distributions of the optimal activated carbon (i.e., SMK-800-30).

**Figure 5 materials-16-06558-f005:**
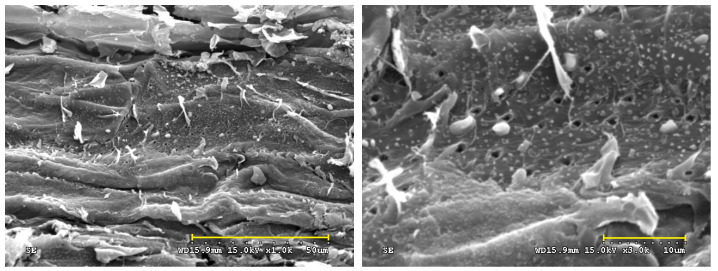
SEM images ((**left**): ×1000; (**right**): ×3000) of resulting activated carbon (SMK-800-30).

### 3.3. Chemical Characteristics of Resulting Activated Carbon Surface

It is well known that most activated carbons contain some oxygen complexes on the surface, which could be derived from their source materials, such as lignocellulosic constituents. The surface oxides of the activated carbons generally pose a polar feature (e.g., hydrophilicity). In this work, these oxides were identified by the analytical instruments of the Fourier Transform infrared spectroscopy (FTIR) and the dispersive X-ray spectroscopy (EDS). Figure 6 depicts various oxygen-containing functional groups in the resulting activated carbon. Based on the general remarks on the functional groups of activated carbon [31,32,33,34], the peak at around 3500 cm^−1^ corresponds to the hydroxyl (O-H) functional group stretching vibration in water molecules (H_2_O). The peak at about 1640 cm^−1^ may be due to C=O groups conjugated with an aromatic ring, indicating the formation of carbonyl-containing groups during carbonization and physical activation. The peak at about 1385 cm^−1^ can be attributed to the oxygen-containing functional groups, for example, C=O and C–O of the carboxylic groups or in-plane vibration of O–H of the carboxylic group. The peak at about 1110 cm^−1^ corresponds to the stretching vibration of the C–O group in alcohol, phenol, ether, or ester. On the other hand, energy dispersive X-ray spectroscopy (EDS) was performed to determine the elemental compositions of the resulting activated carbon semi-quantitatively. As shown in Figure 7, the EDS spectrum of the resulting activated carbon (SMK-800-30) also revealed significant amounts of carbon (C, 74.5 wt%) and oxygen (O, 15.9 wt%) on the sample surface. As compared to the EDS spectrum of the starting precursor (i.e., SMK), these observations were attributed to the gasified release of the oxygen-containing gases (e.g., H_2_O, CO, CO_2_) from its starting lignocellulose during the carbonization/activation process, thus resulting in the oxygen content reduced/remained and the carbon content increased.

On the other hand, the resulting activated carbon also contained inorganic elements (e.g., potassium, magnesium, calcium) derived from starting materials, as shown in Figure 7. These inorganic elements were present in the ash, consisting mainly of alkaline and alkaline-earth-metal oxides such as K_2_O, MgO, and CaO. The functions of these inorganics may increase the hydrophilicity of the activated carbon or the catalytic effects during the activation or steam regeneration step. These inorganic minerals can be removed by washing with a weak acid solution. Due to the feature of the hydrophilicity of the resulting activated carbon mentioned above, this is advantageous for water treatment applications as adsorbents. Furthermore, a preliminary adsorption test for the removal of methylene blue dye (3 ppm) from the solution (2 L) using optimal SMK-based AC (0.6 g) was performed in this work. The results showed a high removal efficiency (>96%) within 60 min, indicating a strong interaction between the cationic adsorbate and the hydrophilic adsorbent.

## 4. Conclusions

In this work, a combination of N_2_-pyrolysis and CO_2_-activation as a single-step process was conducted to produce activated carbon from a novel biomass precursor (*Swietenia macropnylla* King, SMK) seed husk under the combined process conditions of activation temperatures (700–850 °C) and holding times (0–60 min). The texture characteristics were further enhanced at higher activation temperatures from 700 to 800 °C and longer residence times from 0 min to 30 min, but they decreased gradually as the process conditions increased. This result can be attributed to the gasification reaction, causing structural collapse and/or burn-off. We also concluded that the optimal activation conditions for producing the SMK-based activated carbon with microporous features were performed at 800 °C for a holding time of 30 min, resulting in the maximal pore properties (i.e., BET surface area of 966 m^2^/g and total pore volume of 0.43 cm^3^/g). On the other hand, the chemical characteristics of the resulting activated carbon indicated the hydrophilic surface due to the significant oxygen complexes, based on the analyses of the energy dispersive X-ray spectroscopy (EDS) and the Fourier Transform infrared spectroscopy (FTIR).

## Figures and Tables

**Figure 1 materials-16-06558-f001:**
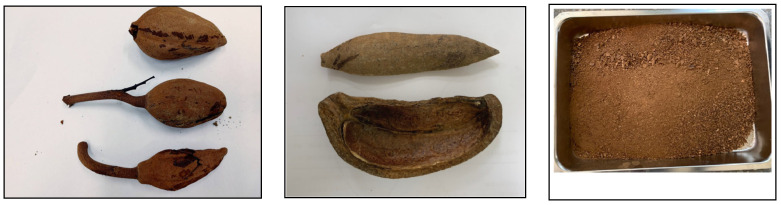
SMK biomass sample (**left**: seeds, **middle**: flaky husk, **right**: fine granule).

**Figure 2 materials-16-06558-f002:**
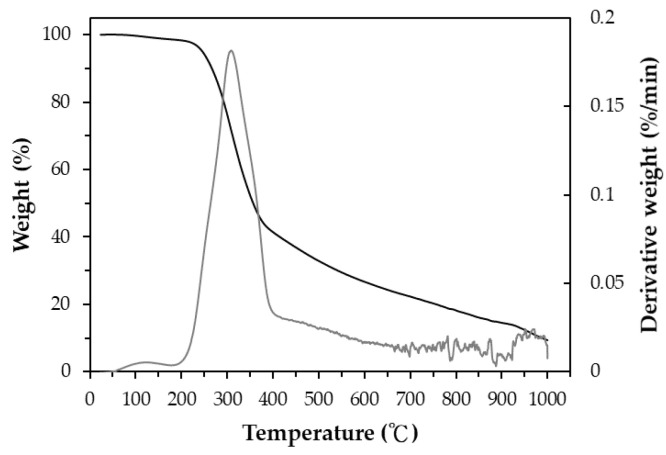
TGA and DTG curves of dried SMK sample (Black line: TGA; grey line: DTG).

**Figure 6 materials-16-06558-f006:**
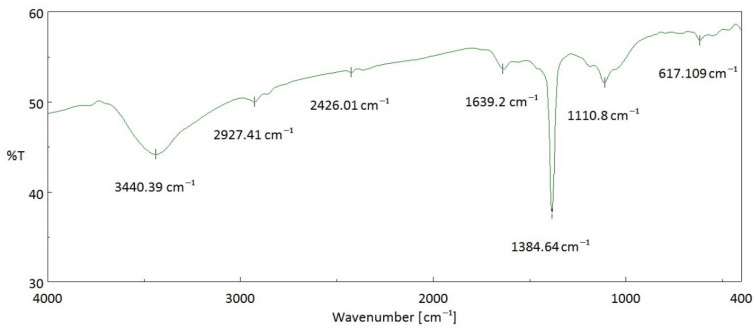
Fourier Transform infrared spectroscopy (FTIR) spectrum of SMK-based activated carbon.

**Figure 7 materials-16-06558-f007:**
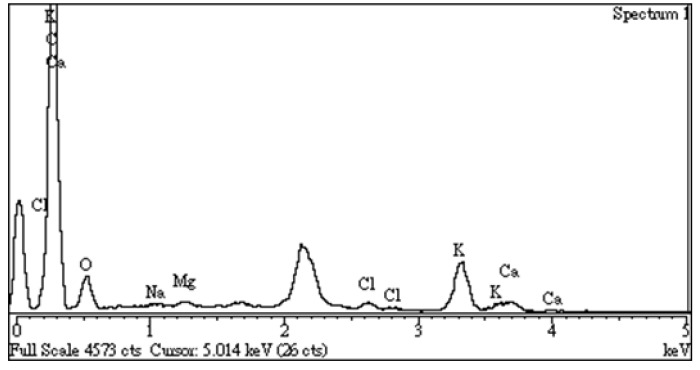
Energy dispersive X-ray spectroscopy (EDS) spectrum of SMK-800-30.

**Table 1 materials-16-06558-t001:** Texture characteristics of SMK-based activated carbon products.

Activated Carbon ^a^	Yield (wt%)	S_BET_ ^b^ (m^2^/g)	S_micro_ ^c^ (m^2^/g)	V_t_ ^d^ (cm^3^/g)	V_micro_ ^c^ (cm^3^/g)	W_p_ ^e^ (nm)
SMK-800-30-Biochar ^f^ SMK-700-30	22.51 26.76	216 ^h^ 311 ^h^	159 241	0.10 0.14	0.08 0.12	4.22 4.18
SMK-750-30	23.87	405 ^h^	277	0.20	0.14	4.52
SMK-800-30	13.39	966	600	0.43	0.30	4.30
SMK-825-30 ^g^	6.01	791	452	0.36	0.23	4.24
SMK-835-30	3.83	475	235	0.21	0.12	3.96
SMK-800-00	24.43	350 ^h^	267	0.15	0.13	4.00
SMK-800-15	19.89	402 ^h^	288	0.19	0.15	4.50
SMK-800-30 ^g^	13.39	966	600	0.43	0.30	4.30
SMK-800-45	7.90	909	547	0.41	0.28	4.32
SMK-800-60	5.41	609	328	0.29	0.17	4.46

^a^ Sample notation indicated the resulting activated carbons produced at the activation temperature of 700–850 °C for holding time of 0–60 min using 5 g SMK. ^b^ BET surface area (S_BET_) was based on relative pressure range of 0.05–0.15 (4–8 points). ^c^ Micropore surface area (S_micro_) and micropore volume (V_micro_) were obtained by using the *t*-plot method. ^d^ Total pore volume (V_t_) was obtained at relative pressure of about 0.995. ^e^ Average pore width (W_p_) for slit pore geometry was roughly calculated from the ratio of the total pore volume (V_t_) and the BET surface area (S_BET_) (i.e., W_p_ = 2 × V_t_/S_BET_). ^f^ As a reference sample. ^g^ Identical. ^h^ The value of parameter C is still less than zero by adjusting the relative pressure range to 0.05–0.15.

## Data Availability

Data are contained within the article.

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
