# Peer review of "Optimization of Physical Activation Process by CO2 for Activated Carbon Preparation from Honduras Mahogany Pod Husk"

_materials, 2023, doi:10.3390/ma16196558_

Round 1
Reviewer 1 Report
The manuscript entitled “Optimization of physical activation process by CO2 for activated carbon preparation from Honduras Mahogany pod husk” not is a relevant contribution for Carbon Capture Science & Technology. From my point of view, this manuscript cannot be considered for publication, because there are many misconceptions. In order to improve the manuscript, the following suggestions/comments/questions should be considered:
- In all article: Change “pore properties” by “texture properties” or “texture characteristics”
- Page 1, line 23: Change “0.429 cm3/g” by “0.43 cm3/g”
- Page 2, lines 62-65: The paragraph “In order to explore the polarity on the surface, the textural and chemical characteristics were obtained from the scanning electron microscopy – energy dispersive X-ray spectroscopy (SEM-EDS) and the Fourier Transform infrared spectroscopy (FTIR)”, none of these experimental techniques allow studying the textural characteristics. This needs to be corrected.
- It is not correct that the SBET values are estimated in the range of 0.05 to 0.3 p/p0, especially if the isotherms of these activated carbons are Type I. It is important to clarify this point, because it is a very common problem that there are errors in the estimation of the SBET. All SBET values have to be estimated according to the Rouquerol criteria described in the IUPAC 2015 report.
- Page 3, lines 123-124: the paragraph “With the model of cylindrical geometry, the average pore diameter (or width) can thus be calculated from the BET surface area and the total pore volume”. The pore geometry of activated carbons, generally is considered as slit pore, and not by cylindrical pore. It is necessary to modified these results.
- In Figure 3, the type of the isotherms must be described and discussed.
- The Y-axis values are not correct, these values, I am sure, which corresponds to volume adsorbed in cm3 STP/g.
- In Table 1: SBET values must be reported without decimal places and pore volume values must be reported with only two decimal places. In addition, the micropore volume (e.g. estimated by Dubinin methods or plot methods) should be added in this table.
- Observing the adsorption-desorption isotherm of N2, the carbon materials are fundamentally microporous and do not present a considerable amount of mesopores. Therefore, the pore size distributions obtained by BJH method using the desorption branch data are not correct. The peak observed (in Figure 4) around 4 nm is an artifact, because there is, in the desorption branch of the isotherms, a little desorption of N2 at aprox. 0.42 of p/p0. The PSD presented in Figure 4 must be deleted and instead the PSD of the micropores should be presented, e.g. those obtained by Horvath-Kawazoe method.
Author Response
Q1. In all article: Change “pore properties” by “texture properties” or “texture characteristics”.
Reply: As suggested by the reviewer, the term “texture characteristics” was used to replace the term “pore properties” in the revised manuscript.
Q2. Page 1, line 23: Change “0.429 cm3/g” by “0.43 cm3/g”.
Reply: As suggested by the reviewer, the value has been rounded to 2 decimal places.
Q3. Page 2, lines 62-65: The paragraph “In order to explore the polarity on the surface, the textural and chemical characteristics were obtained from the scanning electron microscopy – energy dispersive X-ray spectroscopy (SEM-EDS) and the Fourier Transform infrared spectroscopy (FTIR)”, none of these experimental techniques allow studying the textural characteristics. This needs to be corrected.
Reply: As suggested by the reviewer, the description about the textural characteristics has been corrected to show it clear.
“….. The textural characteristics were observed by the scanning electron microscopy (SEM). In order to characterize the oxygen-containing complexes on the surface, the chemical characteristics were obtained from the energy dispersive X-ray spectroscopy (EDS) and the Fourier Transform infrared spectroscopy (FTIR).”
Q4. It is not correct that the SBET values are estimated in the range of 0.05 to 0.3 p/p0, especially if the isotherms of these activated carbons are Type I. It is important to clarify this point, because it is a very common problem that there are errors in the estimation of the SBET. All SBET values have to be estimated according to the Rouquerol criteria described in the IUPAC 2015 report.
Reply: The calculation of the BET surface area depends on the selection of relative pressure (p/p0) range. In this work, the adoption of p/p0 (0.05 to 0.3) range was based on the reference book (Lowell, S.; Shields, J.E.; Thomas, M.A.; Thommes, M. Characterization of Porous Solids and Powders: Surface Area, Pore Size and Density. Springer: Dordrecht, the Netherlands, 2006.), which referred to “The fact that most monolayers are completed in the range 0.05 ≤ p/p0 ≤ 0.3 reflects the value of most C constants.”. Therefore, the range of p/p0.was noted in the determination of the BET surface area. In addition, all correlation coefficients in the calculation of BET surface area are larger than 0.997, meeting one of the Rouquerol criteria (i.e., “A linear fit to the BET transformed data must be obtained.”). Concerning the BET "C" constant positive in the Rouquerol criteria (not always positive by adjusting p/p0 range in this work), different p/p0 ranges (e.g., 0.05-0.10, 0.05-0.15, or 0.05-0.20) for determining the BET surface area must be performed in that way, causing the larger values of BET surface area and the inconsistent comparison.
Q5. In Figure 3, the type of the isotherms must be described and discussed. The Y-axis values are not correct, these values, I am sure, which corresponds to volume adsorbed in cm3 STP/g.
Reply: Thank you for the correction. The units of vertical axes in Figure 3 (and Figure 4) should be “cm3 STP/g”, which has been used in the revised manuscript.
Q6. In Table 1: SBET values must be reported without decimal places and pore volume values must be reported with only two decimal places. In addition, the micropore volume (e.g. estimated by Dubinin methods or plot methods) should be added in this table.
Reply: As suggested by the reviewer, the values of pore volume and surface area have been rounded to 2 decimal places in the revised manuscript. In addition, the micropore volume and micropore surface area (e.g. estimated by t- plot method) have been added to Table 1.
Q7. Observing the adsorption-desorption isotherm of N2, the carbon materials are fundamentally microporous and do not present a considerable amount of mesopores. Therefore, the pore size distributions obtained by BJH method using the desorption branch data are not correct. The peak observed (in Figure 4) around 4 nm is an artifact, because there is, in the desorption branch of the isotherms, a little desorption of N2 at aprox. 0.42 of p/p0. The PSD presented in Figure 4 must be deleted and instead the PSD of the micropores should be presented, e.g. those obtained by Horvath-Kawazoe method.
Reply: We thank the reviewer for the valuable suggestion. In this work, the BJH method was used to evaluate the mesopore (2nm-50nm) size distribution of the resulting activated carbon, not for all pore size distribution. The mesopore size distribution curves in Figure 4 can echo the hysteresis loop in the adsorption-desorption isotherms of N2 (Figure 3). Regarding the pore size distribution of the microporous activated carbon, the Horvath-Kawazoe method is an effective approach for the slit-pore carbon materials. It would be helpful to perform more detailed analysis using the Horvath-Kawazoe method.

Reviewer 2 Report
Comments to the article
Optimization of physical activation process by CO2 for activated carbon preparation from Honduras Mahogany pod husk
This manuscript reports the synthesis of activated carbons using the Honduras Mahogany (Swietenia macropnylla King, SMK) seed husk as a carbon precursor and physical activation to enhance porosity in the resulting carbons. The manuscript is supported by several kinds of experimental data. However, it appears that the experimental methodology and discussion may not offer significant novelty. So, I do not recommend publication of this paper in Materials. Specific comments are as follows:
1) The mass yields of most carbons are too low, applying the high temperature above 800 °C may not be justified in this experiment. I think the experiment should have been planned differently.
2) There is no reference sample, i.e., a carbon prepared without CO2 activation.
3) Authors should measure some specific properties of the synthesized carbons and propose applications.
4) The Introduction section is too brief, the Authors should add some paragraphs as well as cite more reviews related to this research area.
5) Pore volumes should be rounded to 2 decimal places, while values of specific surface area should be rounded to unity.
6) The units of vertical axes in Figure 3 should be (cm3STP·g-1) instead of (mmol/g). Units of vertical axes in Figure 4 are also incorrect.
7) Scale bars are not visible in Figure 5.
8) Authors should remove all the typo and grammatical errors in the manuscript which may interfere with the reading and understanding of the paper.
ok
Author Response
Q1. The mass yields of most carbons are too low, applying the high temperature above 800 °C may not be justified in this experiment. I think the experiment should have been planned differently.
Reply: As listed in Table 1, the mass yields of most carbon products are too low, especially for those produced at higher activation temperatures. In the original experiment design, the activation temperature shall extend to 900°C. However, the mass yields of the resulting activated carbon products produced at above 850°C showed very low values (< 3%). To find the optimal process conditions, two additional activation experiments were performed at 825°C and 835°C for holding 30 min (seen in Table 1).
Q2. There is no reference sample, i.e., a carbon prepared without CO2 activation.
Reply: As suggested by the reviewer, the reference carbon was prepared without CO2 activation at 800°C for holding 30 min. Its yield and pore properties have been added to Table 1, showing the significant differences between the yield and pore properties.
Q3. Authors should measure some specific properties of the synthesized carbons and propose applications.
Reply: As suggested by the reviewer, the description about the methylene blue adsorption of the resulting activated carbon has been added to show its hydrophilic property for water treatment application. This preliminary results have been added to the last paragraph of the Sec. 3.3.
“….. Furthermore, a preliminary adsorption test for removing methylene blue dye (3 ppm) from the solution (2 liters) by using optimal SMK-based AC (0.6 g) was performed in this work. The results showed a high removal efficiency (> 96%) within 60 min, indicating a strong interaction between the cationic adsorbate and the hydrophilic adsorbent.”
Q4. The Introduction section is too brief. the Authors should add some paragraphs as well as cite more reviews related to this research area.
Reply: As pointed out by the reviewer, more detailed description about the research area and the relevant references have been incorporated into the Introduction.
“Due to its excellent texture characteristics, activated carbon has been widely used in a variety of industrial and environmental applications such as gas-phase/liquid-phase adsorbent for purification/remediation [1,2]. Due to the presence of surface oxygen complexes, liquid-phase (mainly water) adsorption for removing organic/inorganic pollutants was more common than gas-phase adsorption. Therefore, activated carbon was sometimes acted as an ion-exchange material. On the other hand, the novel application of activated carbon derived from biomass has been used as electrochemical energy storage devices in recent years [3-5]. In the industrial/commercial production, the precursors for producing activated carbon mainly included coal and hard husk (e.g., coconut shell) [6], but the resulting carbon products are relatively expensive. In this regard, it is necessary to find the new biomass precursor in the production of microporous carbon materials. Therefore, a variety of lignocellulosic residues for producing activated carbon have recently reviewed in the literature [7-10].
Swietenia macropnylla King (SMK) belongs to the family Meliaceae, which is commonly known as mahogany. It is native to the tropical region of America, spreading from southern Mexico to the North of Brazil [11]. In Taiwan, the SMK has been extensively planted in the plain area to exploit it as available wood materials for making furniture and other advanced wood products. Its seeds (or pods) often fall to the ground after ripening or blowing by the wind. Concerning the utilization of SMK seed pod, it can be reused as a source of natural colorants, which may be important dyes in the fabric texture [12]. In addition, the SMK seeds can be collected to be reused as an energy source due to its lignocellulosic constituents [13]. In this work, a novel biomass Honduras Mahogany (Swietenia macropnylla King, SMK) seed husk (pod) was tested as a starting material in the production of activated carbon.”
Q5. Pore volumes should be rounded to 2 decimal places, while values of specific surface area should be rounded to unity.
Reply: As suggested by the reviewer, the values of pore volume and surface area in Table 1 have been rounded to 2 decimal places in the revised manuscript.
Q6. The units of vertical axes in Figure 3 should be (cm3 STP/g) instead of (mmol/g). Units of vertical axes in Figure 4 are also incorrect.
Reply: Thank you for the correction. The units of vertical axes in Figure 3 and Figure 4 should be “cm3 STP/g”, which has been used in the revised manuscript.
Q7. Scale bars are not visible in Figure 5.
Reply: As suggested by the reviewer, the yellow scale bar has been added to make the Figure visible.
Q8. Authors should remove all the typo and grammatical errors in the manuscript which may interfere with the reading and understanding of the paper.
Reply: As suggested by the reviewer, some spelling errors and English syntax have been corrected.

Reviewer 3 Report
Comments: This article reports the optimization of the physical activation process by CO2 for activated carbon preparation from Honduras mahogany pod husk. The structure of the synthesized materials has been characterized by physicochemical characterization. However, authors should address the following comments for its acceptance.
1. Numerous studies on similar materials have been reported in the literature, how your work is different from published work? What is the novelty of the present work? All these points should be explained.
2. Authors need to supply the XRD patterns of prepared materials to determine their phase purity.
3. Author needs to supply the TEM/HRTEM images of the prepared activated carbon to determine their morphology and porous nature.
4. The optimized conditions are unclear. Author needs to explain elaborately.
5. The authors must double-check the whole manuscript to eliminate syntax and format errors.
Minor spell check is required.
Author Response
Q1. Numerous studies on similar materials have been reported in the literature, how your work is different from published work? What is the novelty of the present work? All these points should be explained.
Reply: Indeed, numerous studies on the production of activated carbon (AC) from lignocellulosic material have been reported in the literature. However, this work focused on the use of a novel biomass (i.e., SMK), which was not yet used as a precursor for producing AC in the literature. In addition, the investigation of the optimal process conditions for producing AC was rarely discussed. Therefore, the novelty of the present work will be highlighted in the last paragraph of the Introduction.
Q2. Authors need to supply the XRD patterns of prepared materials to determine their phase purity.
Reply: In general, activated carbon is a non-crystalline material unless it is produced at very high temperature. In this regard, the use of the XRD for identifying the crystalline patterns and the phase purity. The activation temperatures at 700-835°C were performed to produce the AC from low-ash precursor, meaning that this analysis tool was not relevant to the present study.
Q3. Author needs to supply the TEM/HRTEM images of the prepared activated carbon to determine their morphology and porous nature.
Reply: Indeed, the examination of organic/inorganic phases in prepared activated carbon using SEM-EDS can be complemented by using transmission electron microscopy/high-resolution transmission electron microscopy (TEM/HRTEM). As pointed out the Q2, the analysis of TEM/HRTEM is generally used to determine the crystal structure and defective nature of the carbon materials like graphite. It would be helpful to perform more detailed analysis using TEM/HRTEM.
Q4. The optimized conditions are unclear. Author needs to explain elaborately.
Reply: As suggested by the reviewer, a more detailed description about the optimized conditions has been added to explain it elaborately.
“It was found that the mass yields of activated carbon produced at higher activation temperatures and larger holding times decreased continuously, causing the gradual shrinkage and burn-off. Because of more gasification (activation) reactions by CO2 gas at elevated temperatures, the resulting activated carbon product will not be produced or burned out at above 850°C for holding 30 min. Therefore, the lower activation temperatures at 825°C and 835°C were performed to see the variations on the mass yields and texture characteristics. On the other hand, the texture characteristics of activated carbon increased as the activation temperature increased from 700 to 850°C, giving more pore formation and the increment of the mesopore surface area (mesoporosity). Eventually, the effects of activation (or gasification) on the texture characteristics will reach the optimal process conditions where the value of the BET surface area (or total pore volume) will be close to the maximum. As listed in Table 1, the main texture characteristics (BET surface area) increased with increasing activation temperature from 700 to 800°C for holding time of 30 min, but gradually decreased as the temperature increased thereafter from the BET surface area of 797 m2/g (SMK-800-30) to 678 m2/g (SMK-825-30) and 414 m2/g (SMK-835-30). More consistently, the total pore volume indicated a similar trend, suggesting more micropores formed at higher activation temperatures of 700°C to 800°C. Based on these texture characteristics, the activation temperature ranging from 750 to 800°C can be connected with the determining process conditions for the most surface area and pore volume gained in this work.”
Q5. The authors must double-check the whole manuscript to eliminate syntax and format errors (Minor spell check is required.).
Reply: As suggested by the reviewer, some spelling errors and English syntax have been corrected.

Round 2
Reviewer 1 Report
Although there are several points that were considered and modified, there are still serious conceptual problems that are being committed, fundamentally, when interpreting and obtaining the textural properties (specific surface area and pore size distribution). For this reason, I continue to maintain my position that it is an article that should be rejected.
Author Response
Q1. Although there are several points that were considered and modified, there are still serious conceptual problems that are being committed, fundamentally, when interpreting and obtaining the textural properties (specific surface area and pore size distribution).
Reply: We thank the Reviewer for his/her comment on our efforts in doing the revision. Concerning the textural properties (specific surface area and pore size distribution), we further revised the manuscript by marking yellow color based on the original comments. Therefore, a new figure (Figure 5) has been added to depict the micropore size distribution of the optimal activated carbon (i.e., SMK-800-30) using the 2D-NLDFT-HS model. Also, the data on the BET surface area have been corrected to meet the Rouquerol criteria (i.e., the value of the parameter C exceeds zero, denoting the positive y-intercept of the linear region). It should be noted that the selection of relative pressure range has been changed from 0.05-0.3 to 0.05-0.15, ensuring the validity of the BET model analysis for the microporous SMK-based activated carbon.
Reviewer 2 Report
In my opinion, some of the raised important remarks were not correctly addressed, including those of Reviewer 3.
Author Response
Q1. In my opinion, some of the raised important remarks were not correctly addressed, including those of Reviewer 3.
Reply: We thank the Reviewer for his/her positive comment on our efforts in doing the revision. Although the positive comment by Reviewer 3 (“The authors have moderately addressed the issues raised by the reviewers. Hence, the revised version of the manuscript may acceptable to the journal standard.”), we further revised the manuscript by marking yellow color based on the original comments.
Reviewer 3 Report
The authors have moderately addressed the issues raised by the reviewers. Hence, the revised version of the manuscript may acceptable to the journal standard.
Author Response
Q1. The authors have moderately addressed the issues raised by the reviewers. Hence, the revised version of the manuscript may acceptable to the journal standard.
Reply: We thank the Reviewer for his/her positive comment on our efforts in doing the revision.
Round 3
Reviewer 1 Report
The manuscript entitled “Optimization of physical activation process by CO2 for activated carbon preparation from Honduras Mahogany pod husk” not is a relevant contribution for Carbon Capture Science & Technology. From my point of view, this manuscript can be reconsider after major revision, because there are many misconceptions. In order to improve the manuscript, the following suggestions/comments/questions should be considered:
* Observing the adsorption-desorption isotherm of N2, the carbon materials are fundamentally microporous and do not present a considerable amount of mesopores. Therefore, the pore size distributions obtained by BJH method using the desorption branch data ARE NOT CORRECT. The peak observed (in Figure 4) around 4 nm is an artifact, because there is, in the desorption branch of the isotherms, a little desorption of N2 at aprox. 0.42 of p/p0.
* In Table 1: SBET values must be reported without decimal places and pore volume values must be reported with only two decimal places. Additionally, the superscripts in the Table 1 (e.g. (c) and (d)) must be corrected.
* In Table 1: Average pore diameter (Dp) value IS NOT REPRESENTATIVE for these materials. This value should be replaced with Average pore width for slit pore geometry.
Author Response
Q1. Observing the adsorption-desorption isotherm of N2, the carbon materials are fundamentally microporous and do not present a considerable amount of mesopores. Therefore, the pore size distributions obtained by BJH method using the desorption branch data ARE NOT CORRECT. The peak observed (in Figure 4) around 4 nm is an artifact, because there is, in the desorption branch of the isotherms, a little desorption of N2 at approx. 0.42 of p/p0.
Reply: Indeed, the resulting carbon materials are fundamentally microporous and do not present a considerable amount of mesopores. Therefore, this pertinent comment has been added to echo the Figure 4 deleted.
Q2. In Table 1: SBET values must be reported without decimal places and pore volume values must be reported with only two decimal places. Additionally, the superscripts in the Table 1 (e.g. (c) and (d)) must be corrected.
Reply: As suggested by the reviewer, the SBET values have been reported without decimal places. However, the pore volume values were reported with only two decimal places. Concerning the superscripts in the Table 1 (e.g. (c) and (d)), they have been corrected.
Q3. In Table 1: Average pore diameter (Dp) value IS NOT REPRESENTATIVE for these materials. This value should be replaced with Average pore width for slit pore geometry.
Reply: As suggested by the reviewer, the term (“average pore width for slit pore geometry”) has been used in the revised manuscript.
Reviewer 2 Report
Please at least correct before publication:
1. Values of specific surface area should be rounded to unity
2. The units of vertical axes in Figure 4 should be (cm3/g nm) instead of (cm3STP/g)
Author Response
Q1. Values of specific surface area should be rounded to unity.
Reply: As suggested by the reviewer, the values of specific surface area have been rounded to unity.
Q2. The units of vertical axes in Figure 4 should be (cm3/g nm) instead of (cm3STP/g).
Reply: The units of vertical axes in Figure 4 have been corrected to use “(cm3/g nm)”.
Round 4
Reviewer 1 Report
In Table 1: Average pore width for slit pore geometry should be calculated by: wp = 2Vp/SBET
Author Response
Q1. In Table 1: Average pore width for slit pore geometry should be calculated by: Wp = 2Vp/SBET.
Reply: As pointed out by the reviewer, the data of average pore width for slit pore geometry in Table 1 have been corrected by using the equation. Also, the description about the average pore width for slit pore geometry has been properly amended by marking purple color in the revised manuscript.